# Differences in Administration of Methotrexate and Impact on Outcome in Low-Risk Gestational Trophoblastic Neoplasia

**DOI:** 10.3390/cancers14030852

**Published:** 2022-02-08

**Authors:** Emelie Wallin, Isa Niemann, Louise Faaborg, Lars Fokdal, Ulrika Joneborg

**Affiliations:** 1Department of Women’s and Children’s Health, Karolinska Institutet, 171 21 Stockholm, Sweden; emelie.wallin@regionstockholm.se; 2Department of Pelvic Cancer, Karolinska University Hospital, 171 76 Stockholm, Sweden; 3Department of Clinical Medicine, Aarhus University, 8200 Aarhus, Denmark; isa.niemann@dadlnet.dk; 4Department of Obstetrics and Gynecology, Aarhus University Hospital, 8200 Aarhus, Denmark; 5Department of Oncology, Vejle Hospital, 7100 Vejle, Denmark; louise.faaborg.larsen@rsyd.dk; 6Department of Oncology, Aarhus University Hospital, 8200 Aarhus, Denmark; lars.fokdal@auh.rm.dk

**Keywords:** low risk gestational trophoblastic neoplasia, oral methotrexate, methotrexate treatment, methotrexate resistance

## Abstract

**Simple Summary:**

Low-risk gestational trophoblastic neoplasia is a rare but highly curable malignancy. The most common first line treatment is methotrexate, which can be administered in different forms. In order to investigate the impact of route of administration on methotrexate resistance, toxicity demanding treatment switch, complete remission and relapse, we performed an observational study including women with low-risk gestational trophoblastic neoplasia in a population-based setting in Sweden and Denmark. We found that oral compared to intra-muscular administration of methotrexate gives a higher rate of drug resistance, but does not affect rates of complete remission, recurrence or overall survival. Intra-muscular treatment was associated with more toxicity leading to switch of treatment. We conclude that, although a larger proportion of women develop drug resistance, oral methotrexate, which is easy to administer and highly tolerable, could be an option for well-informed and motivated women.

**Abstract:**

Methotrexate (MTX) is frequently used as first-line treatment for low-risk gestational trophoblastic neoplasia (GTN). Intravenous and intramuscular (im) routes of administration are the most common methods, although oral administration is used by some Scandinavian centers. The primary aim of this study was to assess the impact of form of administration (im/oral) on resistance to methotrexate (MTX-R) treatment in low-risk GTN. Secondary aims were time to hCG normalization, rates of toxicity-induced treatment switch, and rates of complete remission and recurrence. In total, 170 women treated at Karolinska University Hospital in Sweden and Aarhus University Hospital in Denmark between 1994 and 2018 were included, of whom 107 were given im and 63 oral MTX. MTX-R developed in 35% and 54% in the im and oral groups, respectively (*p* = 0.01). There was no difference in days to hCG normalization (42 vs. 41 days, *p* = 0.50) for MTX-sensitive women. Toxicity-induced treatment switch was only seen in the im group. Complete remission was obtained in 99.1% and 100% (*p* = 0.44), and recurrence rate within one year was 2.8% and 1.6% (*p* = 0.29). The form of administration of MTX had a significant impact on development of MTX-R and treatment-associated toxicity, but does not affect rates of complete remission, recurrence or survival.

## 1. Introduction

Gestational trophoblastic disease (GTD) is a group of rare but highly curable tumors arising from abnormal placental tissue. It encompasses the pre-malignant forms, complete hydatidiform mole (CHM) and partial hydatidiform mole (PHM), and the malignant counterparts, invasive mole, choriocarcinoma, placental site trophoblastic tumor and epithelioid trophoblastic tumor. The malignant forms can arise from any type of antecedent pregnancy and are collectively called gestational trophoblastic neoplasia (GTN) [1]. Most forms of GTD produce the pregnancy hormone human chorionic gonadotropin (hCG) at levels which correspond to disease volume, making hCG an excellent biomarker for disease progression, treatment response and surveillance. Persistent or increasing levels of hCG indicate malignant transformation and is the basis for the diagnosis of post-molar GTN [2]. Patients with GTN are typically treated with chemotherapy and are stratified into prognostic groups using the International Federation of Gynecology and Obstetrics (FIGO) prognostic scoring system, which is based on the extent and duration of disease, levels of hCG, type of antecedent pregnancy, and extent of previous treatment [3]. Patients scoring 0–6 are likely to respond to single-agent chemotherapy, although the risk of drug resistance rises with increasing risk score. Patients scoring ≥ 7 are at high risk of developing resistance to single-agent therapy and therefore receive combination chemotherapy from the outset [4,5]. Most patients in the low-risk group are cured with single-agent chemotherapy, and the overall cure rate approaches 100% [6]. The most commonly used drugs are methotrexate (MTX) and actinomycin-D (Act-D) in different regimens, with intravenous and intramuscular (im) routes of administration [7,8]. Studies comparing the effectiveness of these drugs are diverging, probably reflecting patient selection criteria, as well as differences in dosage, frequency and route of administration [9]. There is, however, some evidence that Act-D is more likely to be associated with a higher primary cure rate, but also with more adverse events, particularly mucositis, alopecia and dermatological side effects [10,11,12]. Many centers tend to prefer MTX as first-line chemotherapy due to its mild toxicity profile. 

Oral administration of MTX is routinely used in the treatment of rheumatoid arthritis. Studies comparing the bioavailability of MTX when administered orally or parenterally all demonstrate superiority of the parenteral route, and the efficacy and toxicity for MTX appear to be related to the absorbed dose [13,14]. However, this does not suggest that all patients are in need of parenteral MTX, but rather that the choice can be made based on disease severity, compliance and preference. Patients with an inadequate clinical response to oral MTX seem to benefit from a switch to parenteral administration and dose escalation [15]. 

Oral administration of MTX for the treatment of GTN has historically been described in small case series of non-metastatic patients, and in a combat environment, with high cure rates [16,17,18,19]. At Aarhus University Hospital in Aarhus, Denmark, oral MTX is routinely used as first-line treatment for all cases of GTN, irrespective of risk score. In the high-risk group, induction therapy with oral MTX is believed to limit the risk of severe bleeding at start of treatment. In their report of 30 years of treatment, the rate of complete remission with MTX alone was 55% in the low-risk group and none in the high-risk group. More importantly, all MTX-resistant (MTX-R) patients were cured with second- or third-line chemotherapy [20]. No previous study has, however, compared outcomes after im and oral administrations of MTX. 

The primary aim of this study was to compare rates of MTX-R after im and oral routes of administration of MTX as first-line single-agent treatment for patients with low-risk GTN. Secondary aims were to assess time to hCG normalization as an estimate of required length of treatment, rates of toxicity-induced switch of treatment, and rates of recurrence after im and oral MTX.

## 2. Materials and Methods

This was a retrospective observational study including women with low-risk GTN in a population-based setting with centralized cancer care. Treatment for all forms of GTD is publicly available for all residents of Sweden and Denmark, and no privately funded care is available. Karolinska University Hospital in Stockholm, Sweden, was at the time of the study the referral center for all women with GTN in an area of 4.5 million inhabitants. Aarhus University Hospital in Aarhus, Denmark, is the referral center for all women with GTN in an area of 3.1 million inhabitants. 

### 2.1. Patients

All women with a primary diagnosis of low risk GTN (risk score 0–6) according to the FIGO 2000 prognostic scoring system treated at Karolinska University Hospital and Aarhus University Hospital between 1994 and 2018 were identified in local registers. Medical records were screened and data on patient and tumor characteristics were collected. Routes of administration of MTX and data on treatment response, reason for switch of treatment, time to hCG normalization, and overall outcome were registered.

### 2.2. Treatment Protocols

The protocol used for im administration of MTX was the widely used eight-day regimen with MTX 50 mg administered im on days 1, 3, 5 and 7, and folic acid 15 mg orally on days 2, 4, 6 and 8, repeated every 14 days, adding up to 200 mg MTX per cycle [21]. According to the treatment protocol, two cycles were given as consolidation therapy after hCG normalization. This was changed to three cycles in 2012.

The protocol for oral administration of MTX included MTX 10 mg given orally days 1–5 every 2–4 weeks, adding up to 50 mg MTX per cycle. Tablets of 2.5 mg MTX were taken every 6 h for 5 consecutive days, with 1 mL of folic acid administered as mouthwash after every pill taken. The protocol differed during the study period, with four-week intervals being used in the beginning of the study period and three-week intervals in the end of the study period. The two-week interval was used for two patients treated in Sweden. According to the treatment protocol, one cycle was given as consolidation therapy when hCG reached normal levels.

Indications for treatment switch were MTX-R and MTX toxicity.

### 2.3. Outcome Variables

The outcome variables were rate of MTX-R, rates of toxicity-induced treatment switch, time to hCG normalization, and rates of recurrence in patients given im and oral MTX respectively as first-line treatment for low-risk GTN. 

The current definition of MTX-R in both centers is a plateau in hCG (<10%) over two courses or a rise in hCG between courses. However, there was no clear and uniform definition of drug resistance between centers over the whole study period. The definition of MTX-R in this study was therefore a switch to second-line chemotherapy due to presumed drug resistance based on either plateauing or increasing levels of hCG.

Time to hCG normalization was measured in days from start of treatment to first normal hCG.

Recurrence was defined as consistently increasing hCG levels with or without clinical or radiological progression or regrowth of tumor.

### 2.4. Statistical Analysis

Descriptive statistics are presented as numbers and proportions or medians and interquartile range as appropriate. Distributional differences in clinical factors were tested using Fisher’s exact test for categorical variables and the Mann–Whitney U test for continuous variables. The effect of route of administration of MTX and other clinical variables on MTX-R was estimated using logistic regression models. The clinical variables considered were histological subtype (hydatidiform mole, choriocarcinoma, unknown) and risk score (0–4 vs. 5–6). The first model estimated the unadjusted effect of route of drug administration on MTX-R. In the second model, adjustment was made for potential confounders. Results are presented as mean differences and 95% confidence intervals. The significance level was set to 5% and all reported *p*-values are two-sided. Time to hCG normalization was estimated with the Kaplan–Meier method. Statistical analysis was performed using the IBM SPSS for Windows version 25.0 (IBM Corp., Armonk, NY, USA).

## 3. Results

A total of 196 women with low-risk GTN were identified in the 25-year study-period. Five women did not receive MTX as primary treatment and were excluded. For analysis of MTX-R, 21 women not at risk of drug resistance were excluded. Data on included and excluded patients are summarized in Figure 1. Change of treatment due to toxicity was only seen in the im group (7/107, 4%). The documented reasons for terminating MTX were elevated liver enzymes (*n* = 4), pleuritis (*n* = 1), typhlitis (*n* = 1), and neurological symptoms (*n* = 1).

One hundred and seventy patients remained for analysis, of whom 107 were treated with im and 63 with oral MTX. Patient and tumor characteristics are summarized in Table 1. The groups were similar for all clinical variables except for patient age, where women given im treatment were older (32.1 vs. 30.0 years, *p* = 0.048). 

### 3.1. Oncological Outcomes

Table 2 summarizes the oncological outcomes. For MTX-sensitive women. There was no significant difference between the groups in days to hCG normalization (41 and 42 days in the im and po groups, respectively). Rates of complete remission were similar between groups, and despite that, patients in the im group were given one more consolidation course than patients in the oral group (2 vs. 1, *p* < 0.002); the relapse rate was similar. There was an almost 20% increase in drug resistance for patients treated with oral MTX compared to im MTX (54 vs. 35%, *p* = 0.01). Second line chemotherapy included Act-D (*n* = 34/37, 92%) and Act-D with or without MTX (*n* = 32/34, 94%) in the im and oral groups respectively (*p* = 0.71). The remainder received multi-agent chemotherapy with EMA/CO or BEP. Third line treatment was needed in 9/107 (8%) and 6/63 (10%) of women treated with im and oral MTX, respectively (*p* = 0.80), and consisted of EMA/CO in the im group and BEP in the oral group. All women reached complete remission except one woman in the im cohort who died of unrelated causes during second line treatment. 

### 3.2. Risk of Methotrexate Resistance

In the logistic regression analysis of MTX-R, oral treatment was associated with significantly higher odds of drug resistance (OR 2.22: 1.18–4.19), and the increased risk remained significant when adjusted for risk score and histology. A tendency for higher odds of MTX-R was seen in the multivariable analysis when comparing intermediate risk (risk scores 5–6) with low-risk scores (0–4) (OR 2.24: 0.90–5.58) (Table 3).

The Kaplan–Meier estimates comparing days to hCG normalization for patients given im and oral MTX are shown in Figure 2 and Figure 3. There was no significant difference in time to hCG normalization between treatment groups, regardless of treatment response to MTX or treatment switch. In the whole group of low risk GTN, as well as in the MTX-R group, there was a non-significant trend towards longer time to hCG normalization for patients given im treatment. 

## 4. Discussion

This study shows that treatment with oral MTX increases the risk of developing resistance to first line MTX. However, no differences in overall oncological outcomes such as normalization of hCG, rate of complete remission, or rate of recurrence or survival could be demonstrated. Switch of treatment due to drug toxicity was only seen in the im treatment group.

Gestational trophoblastic malignancies are highly responsive to treatment with chemotherapy, and the prognosis, especially in the low-risk group, is excellent [6]. In selecting the most appropriate therapy, minimizing toxicity while preserving efficacy become important issues. The reported rate of complete remission on single-agent MTX varies between 30 and 90% [22]. Predictors of resistance include higher risk scores, higher pre-treatment hCG-levels, and a non-molar antecedent pregnancy or a histological diagnosis of choriocarcinoma [23,24,25,26]. The remission rate in the im group in our study is consistent with the previous literature. There was a clear trend towards an increased risk of resistance in the higher risk scores of 5 and 6; however, development of MTX-R was not significantly associated with increasing risk score, and no association to histological subtype was demonstrated. This could partly be due to low numbers of both high-risk scores and non-molar histological subtypes in this cohort. MTX-R in the oral treatment group is higher than previously reported, both from the previous report from the same center and also compared to earlier case series [16,17,20]. Considering that the treatment interval shortened from four to three weeks during the study period, this is somewhat unexpected in the same patient population. The definition of treatment resistance was, however, not clearly defined, and could have altered over time.

Overall complete response rate in this low-risk population, regardless of route of drug administration and development of drug resistance, was, as expected, almost 100%. Apart from one death due to unrelated causes in the im treatment group, all women were eventually cured. Our results are completely consistent with the available evidence of excellent cure rates after treatment for low-risk GTN [6,7,9]. The relapse rates were low, and similar between groups despite the fact that patients in the im group were given on average one more consolidation course compared to patients in the oral group. Previous reports on GTN patients with relapsing disease after chemotherapy for low-risk disease demonstrate a similar risk of relapse of approximately 2% [27]; however, it has been shown that increasing the number of consolidation cycles decreases the risk of relapse [28]. Based on this evidence, the number of consolidation courses have in recent years been increased in the Swedish population; however, in the present study it seems as if one to two cycles of consolidation chemotherapy may be sufficient in low-risk GTN.

Interestingly, we found that the time to hCG normalization in women sensitive to MTX was equal between the groups. We had expected that the higher dosage and shorter treatment interval of MTX used in the im regimen would induce a quicker tumor response and shorten the time to marker normalization. However, the MTX-sensitive individuals appeared to respond to treatment regardless of the form of administration, and we can only speculate that women who respond completely to oral treatment are highly MTX-sensitive and that the selection is made by this form of treatment. The literature on MTX treatment in rheumatological disorders, mainly rheumatoid arthritis, demonstrate that MTX efficacy is related to dose rather than route of administration, with an approximately linear relationship between dose and effect [14]. Since bioavailability may be limited for oral MTX, a switch to a parenteral route allowing for higher doses may increase efficacy for some patients with rheumatoid arthritis with an inadequate response to oral treatment [15]. This may be an option also for low-risk GTN patients who fail oral MTX to salvage some patients and avoid the toxicity of standard second- and third-line chemotherapy. Drug toxicity is another important aspect of the choice of treatment. Toxicity of MTX, although generally mild, is also related to the absorbed dose [14]. Although drug induced toxicity was not specifically investigated in our study, all seven patients who needed to switch treatment due to MTX toxicity rather than resistance were originally treated with im MTX and thus given higher drug doses. Previous reports on MTX toxicity demonstrate the same frequency of therapy switch [29]. Gastrointestinal toxicity is the most frequently reported side effect of oral MTX, but was not a problem in this study [30]. We also did not find any difference in time to hCG normalization in the group as a whole, regardless of the need for therapy switch. One reason for this may be that patients given oral MTX fail earlier and thus switch to more effective second-line treatment earlier, while patients respond to im MTX for longer, selecting the most MTX resistant patients for second-line treatment. Act-D was used for second-line treatment in both groups in the vast majority of cases, and there was a similar proportion of women needing third-line multi-agent treatment, suggesting that MTX, regardless of route of administration and followed by Act-D when needed, will cure approximately 90% of all women with low-risk GTN. Some of the noted difference in MTX-R between the groups could also be an effect of variance in the definitions of chemotherapy resistance between the groups. Additionally, since women in the im group were given their treatment in a health care facility, while women taking oral medication remained at home, compliance to medication could have been an issue, accounting for part of the increased MTX-R in this group.

In addition to having a more favorable toxicity profile, oral MTX for low-risk GTN was effective in 46% of all patients, is easier to administrate, and is more cost-effective. Oral treatment outside of a hospital setting demands, however, high compliance with the medication. For highly motivated women who are aware of the limitations in treatment efficacy and who can be compliant with treatment and follow-up, oral MTX could be an option to reduce toxicity, reduce time in a health care facility, and avoid absence from work. Oral MTX could also be an option for women who live in areas of the world where lack of access or great distance to hospital care is frequent. Although sequential use of oral and im MTX has never been described for GTN patients, it is a well-established treatment strategy in rheumatology, and could possibly be an option also for patients who fail oral MTX, to reduce the need of standard second-line chemotherapy and associated toxicity. In addition, for MTX-sensitive women, dose reduction of im MTX could be an option to reduce toxicity. Further prospective studies with structured study protocols are warranted to compare the efficacy and dosage of im and oral MTX in the treatment of GTN, and to challenge the idea of sequential treatment in the case of MTX-R. Based on the results, oral MTX could possibly be more widely accepted as an alternative first line route of administration of MTX for selected women with low-risk GTN.

Our study is limited by its retrospective design and unknown confounders not accounted for. Furthermore, a larger sample size could have increased the accuracy of the estimates. Moreover, there were several differences between the centers which could have affected the results. The main limitation was the temporal lack of a clear and uniform definition of MTX-R. This implies that the indication for switch of treatment could have differed between the centers, but also differed over time, possibly over- or underestimating the differences in outcomes between treatment strategies. The change of treatment protocol for oral MTX could also have introduced a time bias, and ideally, the protocol should have remained the same during the whole study period and between centers. The strengths of this study include its population-based setting with complete coverage of the population in question, and data prospectively collected in local registers with complete patient information from both centers.

## 5. Conclusions

The administration form of MTX has a significant impact on development of MTX-R and treatment-associated toxicity, but does not affect the rates of complete remission, recurrence or overall survival. Oral MTX could be an option for highly motivated well-informed women who are expected to comply with treatment.

## Figures and Tables

**Figure 1 cancers-14-00852-f001:**
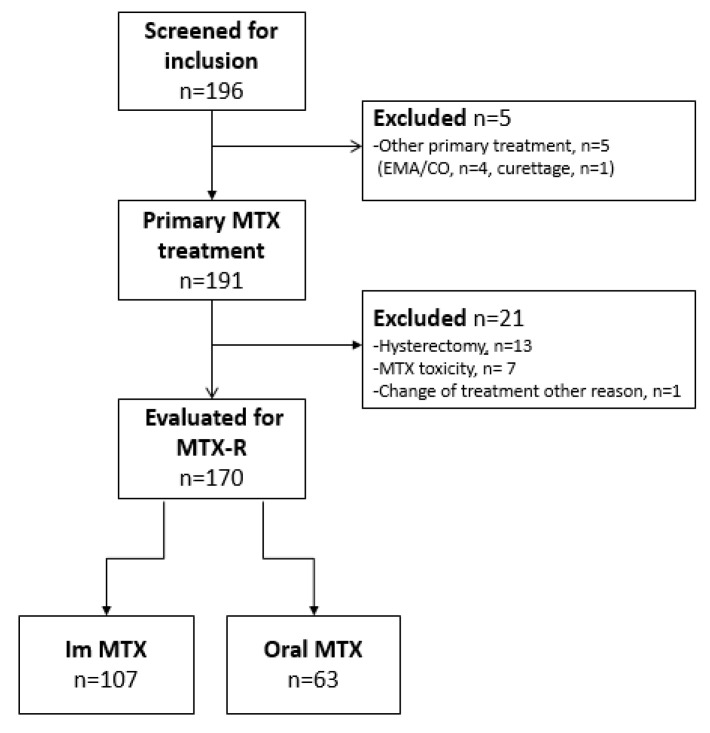
Patient inclusion.

**Figure 2 cancers-14-00852-f002:**
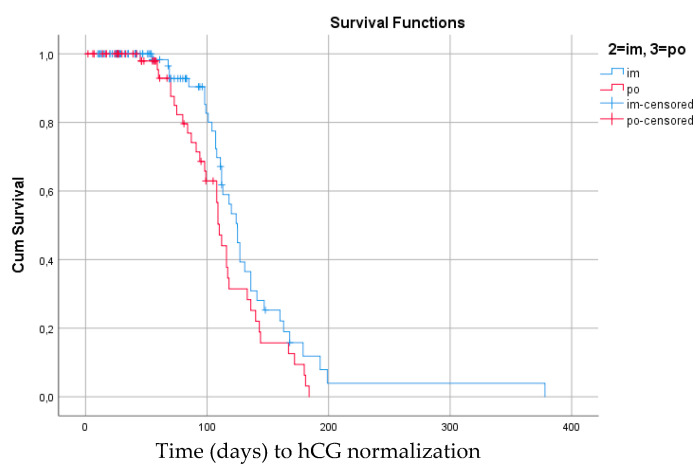
Median time (days) to hCG normalization for all women treated with methotrexate (*n* = 170) as first line treatment for low-risk GTN regardless of drug resistance. Im MTX: 125 days (115.1–134.9) vs. oral MTX: 110 days (104.7–115.3). Log rank 0.067 (95% confidence interval).

**Figure 3 cancers-14-00852-f003:**
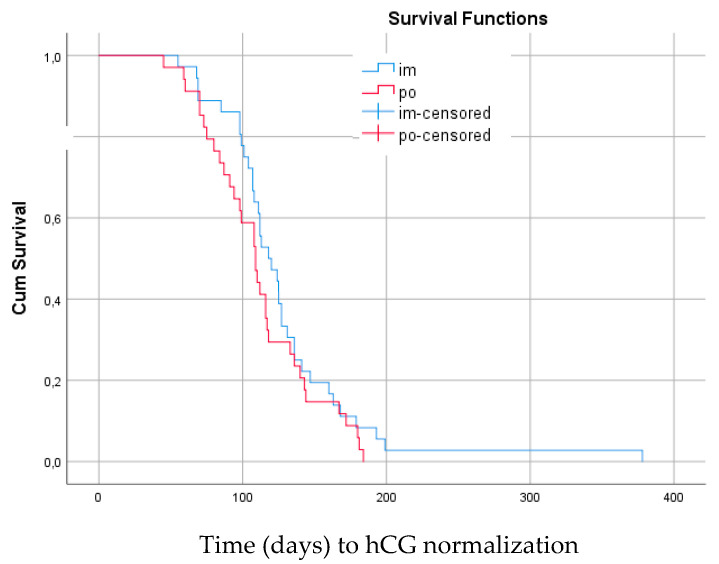
Median time (days) to hCG normalization for women resistant to methotrexate (*n* = 70) as first line treatment for low-risk GTN. Im MTX: 118 days (100.4–135.6) vs. oral MTX: 109 days (96.5–121.6). Log rank 0.22, (95% confidence interval).

**Table 1 cancers-14-00852-t001:** Patient and tumor characteristics.

	IM MTX	ORAL MTX	*p*-Value
	*n* = 107	*n* = 63	
Karolinska, Sweden, *n* = 109	107	2	
Aarhus, Denmark, *n* = 61	0	61	
Age, years, median (IQR)	32.1 (27.6–38.4)	30.0 (26.0–34.0)	<0.05 ^1^
hCG at start of treatment, median (IQR)	6370 (778–20,161)	2486 (230–21,000)	0.29 ^1^
Parity, median (range)	0 (0–3)	1 (0–3)	0.50 ^1^
Risk score, median (range) 5–6 *n*(%)	2 (0–6) 11 (10.3)	2 (0–6) 11 (17.5)	0.45 ^1^ 0.24 ^2^
Histology *n* (%)			
CHM	87 (81.3)	53 (84.1)	1.00 ^2^
PHM	12 (11.2)	4 (6.3)	1.00 ^2^
Choriocarcinoma	3 (2.8)	0 (0)	-
Unknown	4 (3.7)	4 (6.3)	1.00 ^2^
Mole not specified	1 (0.9)	2 (3.2)	1.00 ^2^

^1^ Differences between groups by Mann–Whitney U test, statistical significance at level of *p* < 0.05. ^2^ Fisher’s exact test. Abbreviations: MTX = methotrexate; hCG = human chorionic gonadotropin; CHM = complete hydatidiform mole; PHM = partial hydatidiform mole.

**Table 2 cancers-14-00852-t002:** Oncological outcomes in im vs oral MTX treatment groups.

	IM MTX	ORAL MTX	*p*-Value ^3^
	*n* = 107	*n* = 63	
Days to hCG normalization ^1^	41 (27–69)	42 (22–57.5)	0.5
median (IQR)			
Number of consolidation courses ^1^	2 (0–4)	1 (0–3)	<0.01
median (range)			
MTX-resistance, *n* (%)	37 (34.6)	34 (54)	0.01
Complete remission, *n* (%)	106 (99.1) ^2^	63 (100)	0.44
Relapse after MTX < 1 year, *n* (%)	3 (2.8)	1 (1.6)	0.29
Relapse after MTX > 1 year, *n* (%)	2 (1.9)	0	0.12

^1^ Only MTX-sensitive patients included. ^2^ One patient died of unrelated causes during treatment. ^3^ Differences between groups by Mann–Whitney U test, statistical significance at level of *p* < 0.05. Abbreviations: MTX = methotrexate; hCG = human chorionic gonadotropin; IQR = interquartile range.

**Table 3 cancers-14-00852-t003:** Univariable and multivariable logistic regression analyses of methotrexate resistance as a dependent variable. Data presented as odds ratios and confidence intervals of 95%.

Variables	Category	Univariate			Multivariate		
		OR	*p*-Value	95% CI	OR	*p*-Value	95% CI
Type of treatment	Oral vs. im	2.22	0.01	1.18–4.19	2.21	**0.02**	1.15–4.21
Histological diagnosis	1. Postmolar		0.62			0.60	
	2. Choriocarcinoma	2.89	0.39	0.26–32.57	3.47	0.32	0.30–40.8
	3. Unknown	1.45	0.61	0.35–5.99	1.22	0.79	0.28–5.42
Risk score	5–6 vs. <5	2.24	0.08	0.90–5.58	1.96	0.16	0.77–5.00

Abbreviations: OR = odds ratio; CI = confidential interval; im = intramuscular.

## Data Availability

Data available on request.

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
