# Peer review of "Differences in Administration of Methotrexate and Impact on Outcome in Low-Risk Gestational Trophoblastic Neoplasia"

_cancers, 2022, doi:10.3390/cancers14030852_

Round 1

Reviewer 1 Report

Thank you for asking me to review this paper on assessing the impact of form of administration (im/oral) on resistance to methotrexate (MTX-R) treatment in low-risk GTN. The authors concluded that the form of administration of MTX has a significant impact on development of MTX-R and treatment associated toxicity, but does not affect rates of complete remission, recurrence or survival. Although oral MTX treatment has fewer side effects than im MTX, but it requires the patient to take it at home. Moreover ,oral MTX increases MTX resistance rate by 20%.Thus, management and close follow-up of these patients became more important and somewhat complicated. When the authors recommend oral MTX, this should be stated.

Listed below are some points to help improve this manuscript:

Introduction:

Line 80 “At Aarhus University Hospital in Aarhus, Denmark, oral MTX is rotinely used as first-line treatment for all cases of GTN, irrespective of risk score.” In FIGO Cancer report,MTX can use via intravenously or intramuscularly,can the author explain why they routinely use oral MTX? Are there any other regime used for low-risk GTN patients in these two centers?

Methods:

Line 118. Can the authors state how many consolidation cycles were used? Can they clarify what the process was for selecting second line therapies and what these were.

Results:

Table1: I wonder the portion of FIGO score 5-6 patients in each groups, and whether the portion is different between the two groups.

Can the authors state the diagnosis method of Choriocarcinoma in this table?

Discussion

Line 248: “The literature on MTX treatment in rheumatological disorders, mainly rheumatoid arthritis, demonstrate that MTX efficacy is related to dose rather than route of administration” However,oral MTX is not routinely administered, can the authors explain how to determine the dose and intervals of oral MTX?

Reviewer 2 Report

  1. The authors conducted a multivariate analysis. Why was oral treatment associated with significantly higher odds of MTX resistance?  The authors described that the oral group was not adequately managed by health facilities, but the reason for this was not enough. Is it also necessary to discuss the appropriateness of the dose in oral treatment?
  2. Could you explain the consolidation chemotherapy? Does it mean the treatment after hCG normalization for patients given im and oral MTX? Why does the number of consolidation courses differ significantly between oral and im administration?
